# Zebrafish as a Model of Cardiac Pathology and Toxicity: Spotlight on Uremic Toxins

**DOI:** 10.3390/ijms24065656

**Published:** 2023-03-16

**Authors:** Annapaola Coppola, Patrizia Lombari, Elvira Mazzella, Giovanna Capolongo, Mariadelina Simeoni, Alessandra F. Perna, Diego Ingrosso, Margherita Borriello

**Affiliations:** 1Department of Advanced Medical and Surgical Sciences, University of Campania “Luigi Vanvitelli”, 80138 Naples, Italy; 2Department of Precision Medicine, University of Campania “Luigi Vanvitelli”, 80138 Naples, Italy; 3Department of Translational Medical Science, University of Campania “Luigi Vanvitelli”, 80138 Naples, Italy

**Keywords:** zebrafish, uremic toxin, CKD, cardiac disease, cardiotoxicity

## Abstract

Chronic kidney disease (CKD) is an increasing health care problem. About 10% of the general population is affected by CKD, representing the sixth cause of death in the world. Cardiovascular events are the main mortality cause in CKD, with a cardiovascular risk 10 times higher in these patients than the rate observed in healthy subjects. The gradual decline of the kidney leads to the accumulation of uremic solutes with a negative effect on every organ, especially on the cardiovascular system. Mammalian models, sharing structural and functional similarities with humans, have been widely used to study cardiovascular disease mechanisms and test new therapies, but many of them are rather expensive and difficult to manipulate. Over the last few decades, zebrafish has become a powerful non-mammalian model to study alterations associated with human disease. The high conservation of gene function, low cost, small size, rapid growth, and easiness of genetic manipulation are just some of the features of this experimental model. More specifically, embryonic cardiac development and physiological responses to exposure to numerous toxin substances are similar to those observed in mammals, making zebrafish an ideal model to study cardiac development, toxicity, and cardiovascular disease.

## 1. Introduction

Chronic kidney disease (CKD) is a progressive condition, affecting almost 850 million people worldwide [1]. Defined as a reduction in kidney function, considering the estimated glomerular filtration rate (eGFR) or markers of kidney damage present for at least 3 months, CKD is considered a public health problem, with high rates of mortality due to the association with cardiovascular diseases (CVDs) [2].

The high frequency and combination of CKD and CVDs reflect common mechanisms that join these pathologies. In support of this consideration, for example, is the observation that common traditional risk factors for CKD, such as obesity, systemic hypertension, hyperglycemia, diabetes, and metabolic disorders, are also associated with acute myocardial alteration, atrial fibrillation, stroke, and ischemia [3].

Despite this evidence, the frequency and gravity of CVDs in CKD patients cannot be fully explained with the severity of traditional risk factors, suggesting that CKD-specific pathobiology is involved in the development of CVDs. Numerous studies focused attention on non-traditional risk factors, including inflammation, oxidative stress, mineral-bone disease (CKD-MBD), and the accumulation of uremic toxins. These factors show a clear correlation with different pathologic conditions, such as atherosclerosis, endothelial dysfunction, and vascular calcification, leading to CVDs from mild to moderate stages of CKD [4,5].

Uremia is a complication of CKD, defined as the accumulation of solutes normally cleared by the kidneys [6,7]. The retention becomes more severe in relation to kidney injury, allowing these solutes, defined as uremic toxins, to reach different organs through the bloodstream with a negative impact. Uremic toxins (UTs) can be defined as a residue of organic compounds, which cannot be eliminated from the body. In fact, despite regular dialysis treatment, toxins are not completely removed and become the cause and consequence of CKD [8,9]. Although vascular damage in CKD is exceptionally complex, the cardiotoxicity of UTs has been increasingly demonstrated. UTs contribute to cardiac remodeling and damage, worsening endothelial dysfunction, vascular fibrosis, hypertrophy, oxidative stress, and inflammation with an increased risk of death [7,9,10].

Over the last few decades, zebrafish has become a powerful non-mammalian model to study the alterations associated with human disease. The advantages of using zebrafish include low cost, small size, and rapid growth for statistical evaluation. The popularity of zebrafish can be attributed to their genetic similarity to humans, with about 71% of human genes having orthologs in zebrafish [11]. This provides the possibility of genome manipulations, which allow the creation of zebrafish mutants that reproduce human disease, including cancer [12], diabetes or obesity [13], neurological and neuropsychiatric diseases [14], cardiac pathologies, and toxicity [15].

For many years, other animals, such as rodents, have been considered excellent models for the study of drug toxicity, but are extremely expensive and time-consuming. Nowadays, zebrafish represent a valid alternative vertebrate model to determine developmental toxicity and test compound safety [15].

Specifically, considering the cardiovascular system, zebrafish provides an excellent model for studying heart development and testing drug-induced cardiac toxicity. Due to external fertilization and the transparency during the early stages, zebrafish cardiac growth, as well as cardiac alterations, can be monitored over time [16,17]. In addition, despite the presence of two heart chambers, zebrafish heart morphology and electrical properties, as well as cardiac rate and potential, are analogous to humans. These outstanding properties allow us to study cardiac pathologies, including those that affect the structure of the heart, such as cardiomyopathies, heart failure, myocardial injury, congenital heart disease, and heart regeneration, as well as diseases that affect heart functionality, such as arrhythmia, atrial fibrillation, and sick sinus syndrome [18,19,20].

Cardiotoxicity is defined as the toxicity affecting the heart and cardiac tissues after exposure to certain compounds, e.g., uremic toxins. If severe, cardiotoxicity can be the cause of cardiac alterations, which lead to various cardiomyopathies. Generally, the toxicity reported in zebrafish is representative of other species, including humans [15,21]. In fact, embryonic zebrafish can be considered for studying acute compound-induced toxicity, while adult zebrafish are used as a model for progressive and chronic diseases [15,21].

The aim of this review is to emphasize the importance of zebrafish as an experimental model in the study of cardiovascular alterations and its potential role in valuated cardiotoxicity induced by uremic toxins.

## 2. Zebrafish as a Model at the Forefront of Research

According to the European Commission Directive from 2010, the earliest life stages until the 5-day post-fertilization (dpf) of zebrafish are not regulated as animal studies, offering a valid alternative to animal models [22]. In addition, zebrafish represent a useful model to study human disease because it also follows the principle of the 3R approach:Replacement: Zebrafish is an experimental model to avoid or replace the use of animals. This includes larvae assays, which offer a model of lower potential for pain;Reduction: Its use reduces animal numbers without compromising on animal welfare and is statistically significant in experimental results. Zebrafish larvae, as a first-tier model for toxicity, are used to identify the effect of different compounds with a reduction in the number of animals used in testing;Refine: It allows us to find new ways of minimizing the suffering and pain of animals. In this case, embryos and larvae, with external fertilization and the transparency of the body during development, represent a valid non-invasive observation of toxicities.

Generally, hundreds of embryos can be easily housed in a standard aquarium, allowing not only a reduction of housing space and testing in reasonable sample sizes but also a very high number of replicates at one time. This, together with the ease of making transgenic models, provides the generation of gene expression and cell-specific reporter lines to follow real-time in vivo experiments, useful for the study of physiological and pathological conditions [15,21,22,23]. Like all animal models, zebrafish also present some disadvantages. The lack of maternal-embryo interaction during gestation and the presence, up to 48–72 hpf, of the chorion, may create problems for drug permeability. The extremely rapid development could be a disadvantage, where the screening in zebrafish appears as a running-away train, requiring a perfect experiment setting to have the right exposure time. Many genes in zebrafish are orthologous to humans, but are present in two copies, creating another molecular pathway with different effects than the main one [23].

### 2.1. Zebrafish Life Cycle

Zebrafish (*Danio rerio*) is a tropical freshwater fish, belonging to the minnow family (Cyprinidae). The founding father of zebrafish research was George Streisinger, proving the first method for mutagenesis and the production of zebrafish clones to study the nervous system through genetic analysis [24]. Streisinger and their coworkers also proved techniques for genetic screening in zebrafish [25,26]. From the 1990s, the use of zebrafish as a model organism became increasingly popular, allowing for the partial identification of genes involved in embryogenesis. Over the years, standardized techniques were developed and genetic information was made available in online databases. Since then, more sophisticated techniques have been created, proving that zebrafish have genetic and physiological similarities with humans, including the nervous, digestive, heart and vasculature, kidney, and immune systems [27,28,29,30,31].

The zebrafish life cycle is distinguished by four main stages: the embryo, until 50 h post-fertilization (hpf); larva, until metamorphosis at ~28 dpf; the juvenile, from ~3 months until the end of puberty; and the adult stage. A single breeding couple produces hundreds of eggs, where the external fertilization and the transparency of embryos allow us to explore the morphology and function of developing organs by light microscopy [17]. Embryos develop synchronously and extremely rapidly. During the stage between 10 and 24 hpf, the body of the embryo increases quickly and the sensory tissue start to form. After the formation of the head, fins, circulatory, and other primary systems, at 24 hpf, the heart initiates cardiac contraction. The hatching time can occur between 48 and 72 hpf. During this phase, the morphogenesis is completed and the larvae lose transparency and develop pigmentations. After 5 days, zebrafish begin to be considered experimental animals. At ~6 weeks, zebrafish change their pigment patterns and their fins gain the characteristic of a juvenile appearance. At ~3 months, zebrafish are sexually mature and are considered adults [23,32,33,34]. A detailed scheme of the zebrafish life cycle is reported in Figure 1.

### 2.2. Zebrafish Cardiovascular System: Morphogenesis and Physiology

In zebrafish, the heart is the first organ to form and work during embryogenesis. The spatiotemporal development of the cardiovascular system has been well described over the years. In 2001, Isogai et al., using confocal microscopy, provided a description of cardiac and vascular development within 7 dpf of zebrafish. They also found that, despite the differences in zebrafish vascular anatomy, the principal steps of development showed a strong similarity with respect to other vertebrates [35,36,37,38].

Heart development can be divided into different stages [39,40]. At 5 hpf, the cardiac progenitor cells (CPCs) appear in the lateral marginal zone of the blastula, while the endocardial cells are located toward the lateral margin, without any organization [36,41,42]. During gastrulation, the CPCs move to the posterior half of the anterior lateral plate mesoderm (ALPM) [37,43,44]. From 16 hpf, the CPCs begin the differentiation and fuse into the cardiac disk, with the endocardial cells within the hole at the center, ventricular myocytes at the circumference, and atrial myocytes at the edge of the disc [43,45,46,47,48].

The disc elongation generates a heart tube, that is completely formed at 28 hpf, while the cardiogenic differentiation continues. After the formation, the tube folds toward the right side, becoming the future ventricle, positioned on the opposite side with respect to the atrium, on the left side [49]. At this stage, the irregular cardiomyocyte contractions became coordinated, with an increase of sequential contractions from the atrium and the ventricle, in relation to cardiac development [50]. From 48 hpf, the ventricle and the atrium become morphologically distinguishable as two separate chambers [51]. The heart is still immature, but the major components, such as the bulbous arteriosus, pro-epicardium, and cardiac valves, are formed. At 72 hpf, the trabeculae formations start and rapidly expand in the developing heart, in association with the coronary vasculature and conduction systems [52,53,54]. The pacemaker cells are localized in the inner curvature of the atrium, similar to mammalian pacemaker localization [55]. During the early juvenile stage, the heart continues its remodeling. It rotates and the ventricle, composed of primordial myocardium and spongy trabecular layers, is positioned ventrally to the atrium [34,56,57]. Into adulthood, the ventricle wall includes a compact, highly vascularized myocardium [58].

In contrast to humans, the zebrafish heart is composed of two pumping chambers, with one atrium and one ventricle, contained in a membranous sac, known as the pericardium [59]. Besides the atrium and ventricle, there are two other distinct structures: the sinus venosus and bulbuls arteriosus. Each of these structures has a specific size, shape, and function that contributes to an efficient circulatory system [59].

The deoxygenated blood from the periphery enters the sinus venous, consisting of pacemaker cells and conductive tissue, responsible for the heart contractions [60,61]. From this structure, the blood flow passes into the atrium, a cavitated organ composed of cardiac muscle, collagen, and thin trabeculae [40].

Atrium contractions pump blood into the ventricle through the atrioventricular valve. The ventricle has a pyramidal shape, mainly composed of cardiac muscle and a spongy inner layer with numerous, highly vascularized trabeculae. Because of its structure, the ventricle is the most important component for producing and maintaining blood pressure [58].

Ventricular cardiomyocytes are linked by gap junctions, which generate the synchronism of all cells to the cardiac rhythm [58]. The contractions of the ventricle allow the passage of blood into the bulbus arteriosus via the ventricular bulbar valve. The bulbus arteriosus forms a supplementary pseudo-chamber, with a wall consisting of fibro-elastic tissue and some smooth muscle fibers [62]. During the final stage, from the bulbus arteriosus, blood reaches the gills through the ventral aorta [63]. The scheme of zebrafish heart development is reported in Figure 2.

### 2.3. Zebrafish Vascular System: Vessels Morphogenesis and Structure

Blood vessels are an integral and essential part of all organs, and are important for their morphogenesis and functions. These vessels form a highly ramified network to deliver oxygen, nutrients, hormones, metabolites, and blood cells to all peripheral organs through the arteries, veins, and capillaries, while simultaneously the waste products present in the blood are removed. Zebrafish have a closed circulatory system, and the formation of blood vessels can be divided into two distinct processes: vasculogenesis and angiogenesis [35,64,65,66].

Vasculogenesis is defined as the aggregation of angioblasts into blood vessels. It starts from 14 hpf, when the angioblasts, located in the ventral-lateral mesoderm, migrate toward the embryonic midline, forming the first circulatory loop with the dorsal aorta and the cardinal vein. These two structures form the primitive vascular cord [67]. At 17 hpf, the angioblasts express signs of arterio-venous differentiation and, at 21 hpf, they are localized in the ventral part of the vascular cord, where they start to proliferate under the dorsal aorta, increasing the thickness of the vessels. After the formation of a cord-like structure, the angioblasts start the lumen formation process. This process begins with the hollowing of the cells to form a continuous intracellular lumen; then, the cells migrate in a polarized state, surrounding the primitive lumen [65]. At this point, lumen formation may advance through alternative processes: (i) hollowing, where the cells reach the apical-basal polarity of the cord, and the lumen is formed by membrane separation and fluid influx; (ii) cavitation, where the cells in the middle of the cord go through apoptosis, leaving a luminal space available; and (iii) the wrapping up or budding from an epithelial sheet [68]. Brain vascularization starts at 32 hpf through the formation of premature central arteries and the blood-brain barrier, and the process is completed after the formation of premature central arteries and a blood-brain barrier. At 48 hpf, blood circulates through the aortic arches, entering an anterior and posterior circulatory loop. The anterior vascular loop is connected to the brain vasculature, while the posterior loop relates to the dorsal aorta. The anterior and posterior loops join in the cardinal vein, where the blood comes back to the heart, closing the circulatory loop [35].

Angiogenesis is the process through which new blood vessels are formed from pre-existing ones, expanding the vascular network during organ and tissue formation. Inadequate vessel maintenance or growth led to tissue ischemia, typically related to cancer, inflammatory disorders, and retinopathies [69]. Generally, the dorsal aorta, cardinal vein, and primary cranial vasculature are the first vessels that assemble and establish the major axial vessels [64,70,71]. These vessels, simultaneously, acquire artery and venous identity, not only by their physiological parameters, such as differences in blood flow and pressure, but also in their genetic profiles [72,73].

Sprouting angiogenesis is the predominant mechanism of angiogenetic growth observed in zebrafish. This process can be divided into four steps: tip cell formation, tubule morphogenesis, lumen creation, and the stabilization and maturation of the newly-formed vessels [70,71]. At ~22 hpf, to the right of the dorsal aorta, sprouting angiogenesis initiates in the zebrafish trunk. Each angiogenic sprout is formed by multiple endothelial cells with leading tip cells emerging from the aorta and, by exploratory filopodia, they migrate between the vertical somite boundaries. Subsequently, the stalk cells follow the tip cells, migrating dorsally to form the dorsal longitudinal anastomosing vessel at ~30 hpf. During this process, the vessels become lumenized and a blood flow is established [35].

The vessel walls of zebrafish reflect the same organization in other vertebrates [35]. Moving from the lumen to the periphery, it is possible to distinguish three layers: the tunica intima, media, and adventitia. As opposed to humans, this organization is typically in large vessels and is not found in capillaries and venules. The intima consists of endothelial cells, connected by tight junctions, and an extracellular matrix that forms a basement membrane. The endothelial cell-cell junctions play a crucial role in regulating vascular stability, integrity, and homeostasis, forming a barrier between the blood and tissue. The internal elastic lamina is located between the intima and the media, which, in turn, consist of vascular smooth muscle cells (VSMc) and elastic fibers [35,74]. These structures regulate the internal diameter and distribute the stressing forces uniformly across the vessels, regulating the blood flow and pressure [75,76]. Lastly, the adventitia is the outer layer and consists of connective tissue, made of fibroblast and collagen, with the function of restraining the vessels from excessive extension [77,78]. The differences between fish and mammals include the presence of mural cells in the endothelium and low arterial blood pressure, even if the response of vasculature to the classical vasodilators and vasoconstrictors or drugs is conserved [19,79,80].

## 3. Zebrafish to Study Human Cardiac and Vessel Alterations

The use of zebrafish as a model of cardiovascular human diseases has increased in recent years.

Embryo translucency allows us to easily assess cardiac size and function under a normal stereomicroscope, while adult zebrafish require echocardiography. The utilization of these techniques allows us to evaluate the pumping efficiency of the heart, ventricular and atrium alterations, and the edema associated with cardiac remodeling and heart disease [81,82]. Other techniques, such as time-lapse microscopy, fluorescent microscopy, and micro-computed tomography, may identify changes in cardiac performance in spatiotemporal resolution [81]. Another distinctive feature is the lack of blood circulation in embryos. The oxygen enters the small size of the embryos and reaches all the tissue by passive diffusion. This property offers an incredible advantage for studying severe cardiovascular defects in the early phase of development, in contrast to mammalian embryos, which die rapidly in the presence of important anomalies in the vascular system [18,19,20].

The amenity of zebrafish for genetic manipulation has allowed for the development of several transgenic technologies to study cardiac alterations. Many of these methods include the use of molecular tracers or specific transgenic zebrafish lines to track cardiac structure or cells during the early stages of zebrafish development [83,84].

### 3.1. Genetic Toolbox

The genetic analysis of zebrafish development and physiology was based, in previous years, on mutants identified by genetic screens. Described for the first time in *Arabidopsis thaliana* in 2000, targeting induced local lesions in genomes (TILLING) was the first genetic approach to successfully give germline mutation in the gene of interest in zebrafish. The general approach is very simple: zebrafish are chemically mutagenized, and the genomic DNA is screened for a mutation of interest. In its early form, DNA is amplified, using polymerase chain reaction (PCR), and then digested with CEL1 endonuclease, a plant-specific extracellular glycoprotein that cleaves heteroduplex DNA at all possible single nucleotide mismatches [85]. The fragments with abnormalities are then sequenced and the mutant fish is confirmed if they outcross to a wild-type zebrafish-produced mutant offspring [86,87]. Despite TILLING being a powerful technology to study genetic mutation, it is very expensive and cannot be implemented in most individual laboratories.

Transgenesis consists of the introduction of stable foreign DNA into the genome. By varying the promoter and the gene, the resulting transgenic zebrafish line can be used for a variety of experiments. For example, transgenic lines may express spatial-temporal control of a specific gene, the over-activation of a gene of interest, or express a specific cell type with biosensors, such as fluorophores, particularly useful for drug screening [88].

Nowadays, the application of morpholinos (MOs) is the tool of choice for gene knockdown [89]. MOs are chemically synthesized oligomers, which are typically injected into 1–4-cell-staged embryos [89,90]. MOs bind complementary target mRNA and prevent their splicing and/or translation. Indeed, besides inhibiting splicing, MOs also bind mRNA in the 5′ untranslated region (UTR), blocking translation. The level of knockdown is tested using an antibody to assess the level of the protein of interest, or using a co-injected mRNA or green fluorescence protein (GFP) [91,92,93]. Using MOs is an inexpensive method, and it can represent an important first tool to evaluate gene function. However, MOs present some limitations, such as the possibility that MOs bind other non-specific mRNA or other macromolecules. This situation generates a discrepancy between mutant and morphant phenotypes [36,94].

Two efficient alternatives for reverse genetics in zebrafish are represented by transcription activator-like effector nucleases (TALENs) and the CRISPR/Cas9 system.

TALENs are chimeric proteins, including the endonuclease domain of a FOK I restriction enzyme and a site-specific DNA-binding domain derived from the transcription activator-like effectors (TALEs) of plant pathogens, such as *Xanthomonas*. TALENs consist of multiple 33–35 amino acids containing repeat domains, each recognizing a single base pair. The methods for assembling TALENs involve the use of a restriction enzyme and plasmid [95,96]. The CRISPR/Cas9 system is an adaptative defense mechanism found in archaea and bacteria, considered an adaptive immune strategy for degrading foreign DNA. The CRISPR/Cas9 system is especially useful as a reverse genetic tool because it requires only the enzyme Cas9 and a single guide RNA, which is engineered through the standard DNA cloning technique, to recognize a target site of 20 nucleotides. This approach is efficient for creating mutant-specific loci in zebrafish or for generating mosaic fishes with different mutations [97,98].

### 3.2. Genetic Cardiac Alterations

In zebrafish, heart and vascular alterations are caused by the same underlying molecular regulation and cellular mechanisms present in humans [35].

Congenital heart defects (CHD) are the most common birth defects and comprise various diseases, including septal defects or valve and chamber malformations [99]. CHD can be caused by both genetic and non-genetic factors. Utilizing zebrafish as a model and combining transgenic reporters, via CRISPR/Cas9, different studies have performed functional testing of candidate genes associated with CHD. The results obtained showed that genes encoding transcription factors, such as GATA4/5/6 [100,101], NKX2.5 [102], and HAND2 [103,104], or cell signaling molecules, such as NOTCH1 [105,106] and SMAD6 [107,108] cause CHD in humans and play an important role for heart development in zebrafish. In addition, genes associated with cardiomyocyte function, which cause CHD in zebrafish include *MYH6* [109], *ACTC* [110], *TITIN* [111], and *KCNH2* [112].

Non-genetic factors that cause CHD in zebrafish include hemodynamic forces and cardiotoxic agents. The alterations in hemodynamic forces induce the disruption of normal blow flow, typically involved in the cellular response to shear stress, stretch, and chamber pressure, which culminate in cardiac malformation [113,114].

Cardiomyopathies represent any disease of the heart muscle, typically due to alterations in the cardiomyocyte contractile capacity. Cardiomyopathies comprise three main classes: (i) dilated cardiomyopathy, characterized by an enlarged ventricle, with thinning of the myocardial wall and a reduction of cardiac output; (ii) hypertrophic cardiomyopathy, where the thinning affects the ventricle wall; and (iii) arrhythmogenic cardiomyopathy, typically associated with cardiomyocyte degeneration and the replacement with fibrofatty scar tissue, leading to heart remodeling [115]. All these mechanisms alter the normal cardiac function and, in the end, may culminate in heart failure. Hallmarks of cardiomyopathies and heart failure are conserved in zebrafish carrying mutations in genes, e.g., *MYL4* [116], *TITIN* [117], and *SYNPO2L* [118].

Conduction disorders (CDs) include many pathologies, which depend on the alterations of generation or the propagation of cardiac electrical impulses. Typically, in adult zebrafish, the action potential of cardiomyocytes is similar to humans, in that both have a long plateau phase [119,120]. Additionally, the major inward and outward current systems are qualitatively similar in zebrafish and human hearts [121].

Genetic manipulation in zebrafish has also allowed for the study of three conduction defects, such as long QT syndrome, atrial fibrillation, and sick sinus syndrome.

Long QT syndrome (LQTS) is characterized by prolonged myocardial repolarization time, where ∼40% of cases are caused by defects in the potassium channels (Kv11.1), encoded by the *KCNH2* gene. KCNH2 zebrafish mutants recapitulate the human LQTS phenotype that is manifested by action potential prolongation and the prolonged QT interval [112,122].

Atrial fibrillation (AF) is associated with irregular atrial pacing and an increased heart rate, caused by dysregulation in the PITX2c gene. PITX2c zebrafish mutants show an altered cardiac function with increased atrial fibrosis [123].

Sick sinus syndrome (SSS) is defined by an irregular heart rate with arrhythmia as a consequence. The mutation in the *GNB5* gene appears to be involved only in the parasympathetic control of the heart rate, which translates to extreme bradycardia in mutants [124].

### 3.3. Vessel Alterations

The creation of stable transgenic lines with vascular-specific phenotypes allows us to investigate, in great detail, alterations in blood vessels at both the anatomical and cellular points, with relative molecular players involved. In zebrafish, the most important genes related to the different vessel alterations have been found, including endothelial dysfunction, atherosclerosis, and vascular calcification [125].

Endothelial dysfunction is the first pathological event of vascular damage and has been associated with a host of diseases, including atherosclerosis, vascular calcification, hypertension, coronary disease, diabetes, and heart and renal failure [126]. Atherosclerosis is a chronic inflammatory disease, arising from the buildup of fats, cholesterol, and other substances in the artery walls, which form plaques resulting in lumen occlusion [127]. Zebrafish is a useful model for studying atherosclerosis and hyperlipidemia. Interestingly, zebrafish’s high cholesterol diet can replicate some processes of the early phase of atherosclerosis, such as hypercholesterolemia, LDL oxidation, and vascular lipid accumulation [128,129]. A recent study demonstrated that, with a high cholesterol diet from 5 to 10 dpf, zebrafish show an increase in the signs of inflammations, such as TNF-α and IL-1β, and a decrease in anti-inflammatory markers before lipid deposition in the endothelium [130]. In addition, although the plasma lipid profiles differ between zebrafish and mammals, the lipid metabolism is highly close, showing the conserved genes involved in lipoprotein and lipid metabolism in zebrafish, such as *APOB*, *APOE*, *APOA1*, *LDLR*, *APOC2*, *LPL*, *LCAT*, and *CETP* [131,132]. Several studies have used mutant zebrafish to discover implicated genes in hyperlipidemia and atherosclerosis. For example, Apoc2-deficient zebrafish serves as dyslipidemia and hyperlipidemia models [133]. The LXL mutant shows an increase in LDL, developing severe hypercholesterolemia and hepatic steatosis, especially useful for screening LXL agonists to suppress dyslipidemia and atherosclerosis [134]. The DLR mutants develop moderate hypercholesterolemia, with lipid accumulation in the blood vessels, useful for screening drugs for hypercholesterolemia [135]. In addition to lipid metabolism, the zebrafish alteration in blood flow induces a low or high oscillatory wall shear stress, with an important role in atherosclerosis development and endothelial injury [136].

Vascular calcification (VC) is a pathological event caused by the unusual deposition of minerals in the vascular system. Over the years, it has been demonstrated that VC is an active and regulated process, similar to bone mineralization [136]. Ions, mainly calcium and phosphate, deposited as hydroxyapatite crystals in the vascular walls, increase the thickness and stiffness of vessels, commonly associated with inflammation, atherosclerosis, diabetes, and CKD [137,138].

In zebrafish, pathological mineralization can affect different soft tissues, including the skin, kidneys, cartilage, tendons, eyes, and vasculature, resulting in important morbidity and mortality issues [139]. Considering the remodeling of scales, zebrafish have been proposed as a model for osteogenesis, bone metabolism, and remodeling where the cells, matrix protein, and molecular pathways, such as NOTCH, Wnt, TGF beta, and BMP, are very close to mammals [140]. Singh et al. showed that a loss of α-Klotho, a protective factor that regulates mineral homeostasis in mammals, results in reduced zebrafish lifespans and vascular calcification in the bulbus arteriosus. In addition, the calcification is associated with the ectopic activation of osteoblast differentiation (e.g., ectonucleoside triphosphate diphosphohydrolase 5a, RUNX2b, CLEC11a, and ITGA11a), suggesting that the loss of α-Klotho is related to aging, giving a rise to ectopic calcification [141,142].

In addition to vascular calcification, different studies have also investigated diseases characterized by an ectopic multisystem mineral disorder. For example, the *ABBC6* gene is a member of the ATP-binding cassette transporter family, related to pseudoxanthoma elasticum (PXE), a pathology caused by a mutation in the *ABCC6* and *ENPP1* genes [143]. This gene has two orthologs in zebrafish: *ABBC6A* and *ABCC6BMO*. Sun et al., using CRISPR/Cas9, created different ABCC6A mutants, with a high gene expression in the eyes, heart, and intestines of young adult fishes, which represent a valid strategy for drug discovery against ectopic calcification [143]. Moreover, the *ENPP1* gene, a member of ecto-nucleotide pyrophosphatase/phosphodiesterase implicated in the regulation of pyrophosphate levels, is associated not only with PXE but also with the generalized arterial calcification of infancy (GACI) [144]. To investigate these pathologies, Apschner et al. utilized zebrafish ENPP1 mutants, also called dragonfish. The zebrafish mutants showed generalized calcification in the cartilage, skin, heart, intracranial space, and the notochord sheet [145].

## 4. Uremic Toxins

In the setting of CKD, many compounds, known as the so-called “uremic retention solutes” or “uremic toxins (UTs)”, may accumulate and exert their uremic effects on various systems. Typically, UTs are defined as residues of organic compounds, which cannot be eliminated from the body and, through the bloodstream, reach different organs promoting several functional changes [7,8]. UTs are defined and classified according to Massry/Chock’s requirements, as shown in Table 1.

Considering the importance of the study of UTs, in the 2000s, the Vanholder group was the early pioneer that created the European Working Group on Uremic Toxins (EUTox). EUTox aims to identify and characterize UTs, as well as develop new therapeutic approaches, for the treatment of CKD.

In addition to Massry/Chock’s requirements, UTs are also classified according to Vanholder et al., as shown in Table 2 [8].

Each UT has a specific mechanism of action, which may affect only one or a few systems. Most UTs have an effect on multiple systems, interfering with important cellular mechanisms, such as DNA methylation and repair, oxidative stress, inflammation, and protein binding, with alterations in structure or function as a consequence [146,147,148].

The accumulation of UTs in the blood vessels is related to a variety of vascular alterations, such as the acceleration of atherosclerosis, arterial stiffness, vascular calcification, and intimal hyperplasia [7,10,146,147,148]. Studies have shown that UTs deregulate endothelial functions with structural damage, inflammation, and impaired endothelium-dependent vasodilatation [149,150]. UTs can compromise cell-to-cell communication, disrupting cell junction, the integrity of the endothelial barrier [151], and inducing cytoskeleton remodeling [152], with an increase in vascular permeability and vessel leakage [153]. Endothelial cells play an important role in the expression of proinflammatory cytokines, chemokines, and adhesion molecules, which are extremely elevated in CKD patients, particularly during more advanced stages [154]. UT cardiotoxicity has increased in recent years, showing that these solutes are not only a consequence but also a driver of CKD progression [10,146].

In the next sub-paragraphs, the relationship between the various groups of UTs and CVDs is described.

### 4.1. Free Water-Soluble Low-Molecular Weight Compounds

Free water-soluble low-molecular weight compounds never bind to proteins and most of them are considered inert. The most notable compounds of this group involved in cardiovascular alterations are creatinine, urea, uric acid, and inorganic phosphorous. Other compounds include guanidine and guanidinosuccinic acid, which have a role in vessel damage [155].

Creatinine and urea are typically used as a marker to identify renal function [156]. These compounds are related to cardiomyocyte contractile injury, with an increase in cardiac oxygen consumption, by lowering norepinephrine and causing insulin resistance [9]. Evidence indicates that urea, in CKD, exhibits its toxic effects through carbamylation, a post-transcriptional modification of proteins, with an increased risk of adverse cardiovascular outcomes, such as sudden cardiac death and heart failure [157].

Uric acid is the end-product of purine metabolism in humans. Pathological hyperuricemia is a prognostic marker of cardiovascular mortality in patients with CKD, where elevated serum uric acid may be related to kidney injury [158]. Uric acid levels are associated with an increase in hypertension, heart failure, and atrial fibrillation [157,159,160]. It is also known that uric acid induces inflammation in vascular endothelial and intracellular oxidative stress, leading to endothelial dysfunction [161]. Despite numerous investigations, the mechanism that uric acid contributes to these alterations is still controversial, but the uric acid lowering treatment has consistently improved cardiac prognosis [162].

Trimethylamine N-oxide (TMAO) is an intestinal microbiota-dependent metabolite of dietary choline, phosphatidylcholine, L-carnitine, and betaine [163]. TMAO has been reported to be strongly linked to renal function and increased cardiovascular events in the general population and CKD patients [164]. In particular, the elevation of TMAO levels in CKD is related to the decrease in GFR, increasing renal dysfunction and tubulointerstitial fibrosis in a dose-dependent manner [165].

Inorganic phosphorous (Pi) is approximately 85% of Pi in the bone, 10–15% in soft tissues, and less than 1% in the extra-cellular fluid. Normal levels of Pi are the result of the balance between the intestinal absorption of Pi from the diet, the storage and release of Pi in the bone, and the extraction of Pi through the urine. About 60–70% of dietary Pi, this Pi is absorbed in the intestine via an active sodium/phosphate (Na/Pi) cotransport [166]. Na/Pi cotransport is positively regulated by 1,25-dihydroxyvitamin D [166] and negatively regulated by the parathyroid hormone (PTH) [167]. PTH stimulates bone resorption, enhancing the level of serum Pi. Fibroblast Growth Factor 23 (FGF-23), a bone-derived hormone mainly produced by osteocytes and osteoblasts, inhibits renal phosphate reabsorption and calcitriol synthesis through the inhibition of 1-α-hydroxylase, increasing Pi excretion and decreasing Pi reabsorption in the intestine [167]. High serum Pi levels are associated with cardiovascular disease among patients with or without kidney disease, hypertrophy [168], coronary disease [169], heart failure [170], and VC [171].

### 4.2. Medium Uremic Compounds

Medium uremic toxins have a molecular weight >500 Da, including β2-microglobulin and pro-inflammatory molecules, such as IL-6, β-trace protein (BTP), pentraxin-3, and PTH [172]. Despite some compounds being removed by the dialysis membrane, various toxins contribute to the high mortality and morbidity observed in CKD patients [173].

*β2-microglobulin* is a ubiquitously expressed protein and a component of the major histocompatibility (MHC) class I molecule family. It is produced at a constant rate and freely filtered by the kidney, commonly used as a representative marker of medium molecule retention and removal [174]. β-2-microglobulin appears to be associated with CHD and stroke [175].

*FGF-23* and *PTH* are the key players that have adverse effects on the cardiovascular system. It is important to note that these toxins are naturally synthesized by the organism, and are essential to preserving mineral balance. Elevated plasmatic FGF-23 causes cardiovascular complications since it can promote cardiac hypertrophy development, and cardiac fibrosis and dysfunction [176,177].

*PTH* is a 94 kDa molecular weight peptide, synthesized in the parathyroid gland and stored in secretory granules for release, in response to a reduction in ionized calcium levels. PTH has a double effect: It increases calcium reabsorption, activating calcium channels in the apical membrane of distal tubules [178], and, at the same time, it is responsible for the inhibition of Pi reabsorption in renal proximal tubular cells, and upregulates the 1α-hydroxylase gene, capable of the conversion of 25-hydroxyvitamin D to the active metabolite of vitamin D [1,25-dihydroxyvitamin D (1,25[OH]2D3)] [179]. Recent clinical and molecular research has shown that PTH affects the heart and vasculature. Typically, disorders of PTH levels induce hypertension [180], arrhythmias [181], heart failure [182], and calcific disease [183], which translate into increased cardiac morbidity and mortality.

### 4.3. Protein-Bound Compounds

Protein-bound compounds generated from renal impairment are associated with different alterations, due to albumin-binding properties, making these solutes non-dialysable. Among the protein-bound compounds, Indoxyl Sulfate (IS), P-Cresol (PC), homocysteine (Hcy), and Advanced Glycation end Products (AGEs) have been most extensively investigated in the context of cardiovascular disease.

*IS* is a small solute of 213 Da, formed by host modification of intestinal bacteria, which metabolizes the amino acid tryptophan to indole and is then conjugated with sulfate in the liver. Typically, in the blood, IS binds to albumin to be excreted primarily by proximal tubular secretion and then by glomerular filtration [9], reaching very high clearance, a function that is not replicated by dialysis. IS retained in CKD is associated with several harmful effects on the cardiovascular system, such as endothelial dysfunction, hyperplasia of VSMCs, vascular calcification, and an increase in atherosclerosis in men [184,185]. Studies evaluating the specific toxicity of IS on endothelium demonstrated that endothelial progenitor cells are inversely correlated with IS plasma levels [186,187].

*PC* has a molecular weight of 108 Da, generated from the metabolism of tyrosine and phenylalanine, metabolized by intestinal bacteria [172,188] that process PC through sulfation and glucuronidation. This process generates two compounds: p-Cresyl sulfate (PCS) and p-cresyl-glucoronate, with different impacts on CKD patients [189]. In the human body, PC is present in small concentrations, and it is not detected in CKD. However, about 95% of PCS binds albumin and it is typically found elevated in uremic patients. Some have identified an association between PCS and CVs in stages 3–5 of CKD [190]. The increase in PCS is related to elevated cytokines, pro-inflammatory genes, the complement system and oxidative stress in VSMCs, and human umbilical veins’ endothelial cells (HUVECs), contributing to endothelial dysfunction [191].

*Hcy* is a sulfur amino acid, with a metabolism related to methionine. An increased Hcy, a condition called hyperhomocisteinemia (HHcy), is diagnosticated when the Hcy level in the blood is more than 10 μmol/L in women and more than 12 μmol/L in men [192]. HHcy can be mild (12–15 μmol/L), moderate (16–30 μmol/L), middle (31–100 μmol/L), or severe (more than 100 μmol/L) [193]. A mild HHcy has been recognized as a cardiovascular risk factor in the general population, and in several acquired conditions, such as CKD, where the Hcy levels range from moderate to intermediate. Principally, the Hcy levels can be increased by the defective metabolism of Met, due to either a mutation in the gene coding for the enzyme of the Hcy metabolism [192,194,195] or deficiencies in vitamin cofactors, such as vitamin B6 and B12, involved in its metabolism [192,195]. The Hcy hypothesis as an independent cardiovascular risk factor is supported by the fact that subjects with an alteration in the enzymatic pathway of sulfur amino acid metabolism have a higher level of Hcy than the general population, associated with a faster progression of arteriosclerosis, thrombosis, and inflammation [196]. In CKD, plasma Hcy levels rise when the GFR is reduced by 50%, and in uremia most patients are hyperhomocysteinemic. In addition, dialysis can only partially remove Hcy because of its protein-bound character. The consequences of protein homocysteinylation are protein damage with altered electrophoretic mobility and loss of enzymatic activity (with protein denaturation) in several model systems [197]. Evidence has demonstrated that Hcy has a toxin action. Specifically, Hcy can be harmful to cells because it induces oxidative stress through the production of reactive oxygen species (ROS) and binding nitric oxide [198]. Moreover, Hcy retention leads to the intracellular accumulation of S-adenosylhomocysteine (AdoHcy), the immediate Hcy precursor, a toxic compound, and a powerful competitive inhibitor of biological transmethylations [199]. Macromolecule hypomethylation is a common feature in CKD with possible functional consequences, such as the disruption of DNA methylation patterns with an inappropriate gene expression and the promotion of disease or the increase in protein damage (above proteins containing the L-isoaspartyl residues) [200].

Several other compounds, strictly related to Hcy, have been found elevated in CKD patients and are now under scrutiny as potential uremic toxins, with harmful cardiovascular effects [201,202,203]. In recent years, in undergoing haemodialysis patients, high concentrations of the sulfur amino acid derivative *lanthionine* have been found to increase by about two orders of magnitude than in normal subjects [202,203]. The increase in its concentration depends on reduced clearance but the high level can be dependent on the production of intestinal bacteria in the synthesis of lanthibiotics, a group of post-translationally modified peptides containing unusual amino acids, such as lanthionine residues. It is well known that the CKD microbiota is altered, and it is possible that the increase in blood lanthionine could be linked to this production and subsequent absorption [204].

*AGEs* are a chemically heterogeneous group of compounds that result from the non-enzymatic glycation reaction between reducing sugars (and their derivatives) and amino-containing biomolecules, such as proteins [205]. The damaging effects of AGEs may be due in part to the direct modification and loss of function of those substrates that are involved in AGE formation, such as matrix proteins, and in part to the activation of a pro-inflammatory response following the activation of the membrane-bound AGE receptor (RAGE) [206,207]. The accumulation of AGE participates in renal filtration alteration and glomerulopathy [5,7]. Animal studies supported the involvement of AGE in nephropathy, showing a thickening of the basement membrane and expansion of the mesangial layer after an AGE injection. The links between AGEs and CVD in CKD are multiple. AGEs are involved in arterial stiffness, resulting from non-enzymatic protein glycation, to form irreversible cross-links between long-lived proteins, such as collagen and elastin. The AGEs-linked extracellular matrix is stiffer and less susceptible to hydrolytic turnover, resulting in the accumulation of structurally inadequate matrix molecules. Importantly, experimental studies also showed that the inhibition or breaking of AGEs prevents cardiac hypertrophy and arterial stiffness, and may restore cardiac function [207]. The AGE cross-link breaker improves arterial compliance in humans, as well as the aortic stiffness that is commonly observed as a pathophysiological aspect of large-artery damage. Moreover, the irreversible accumulation of AGEs in vessels, due to their exposure to oxidative stress during the progression of CKD, results in an increase in arterial stiffness and high risks of cardiovascular events in ESRD patients [208]. In addition, AGEs, interacting with RAGE, participate in endothelial damage, frequently associated with CVD [205].

## 5. Zebrafish as a Model to Study Uremic Cardiotoxicity

Cardiotoxicity is defined as the toxicity that affects the heart and other cardiac tissues after exposure to certain compounds. Typically, numerous published studies have used zebrafish embryos, as experimental models, to assess the impact of different compounds, including environmental pollutants, nanoparticles, alcohols, and drugs, such as cocaine and cigarette smoke, on the cardiovascular system [196]. Interestingly, zebrafish have been used to evaluate the cardiotoxicity induced by drugs in different categories, such as anthracyclines (ANTs), cancer drugs, antiarrhythmics, anticonvulsants, and beta-blocker classes [209,210].

Over the years, different software has been developed, allowing an automatic recording of different zebrafish parameters that are useful to evaluate the toxicity induced. Examples are represented by the ZebraBox/ZebraLab and Danio Visio systems. These software programs allow an automated observation of multiple zebrafish larvae and embryos, at the same time, evaluating parameters related to cardiac and behavioral analysis, drug screening and development, circadian rhythmicity, light response, and muscle disorder and recovery [211,212,213]. Despite the numerous signs of progress related to the development of tools for recording and analyzing various zebrafish parameters, only in recent years have zebrafish embryos been used as a model to evaluate uremic toxicity. The first work in this field was provided by Berman et al. in 2013 [214].

This work employs an animal model to link complement activity to uremic toxicity [214]. The complement system is a complex cascade of over 30 proteins that together form part of the innate immune system [215]. The complement is a relatively well-described uremic solute, and its high concentration in uremic patients contributes to vascular diseases [216]. In this study, the zebrafish embryos at 24 hpf were exposed to uremic serum fractionated at 50 kD, obtained from stable end-stage renal disease patients at the pre-dialysis phase. After treatment, at 7 h, the embryos showed a reduction in viability compared to embryos similarly exposed to the control serum. Moreover, the toxicity was strongly associated with the >50 kD component as opposed to the <50 kD component, suggesting the toxic molecule was either larger than 50 kD or protein bound. To investigate the complement toxicity, the authors used specific inhibitors of a cascade, showing a reduction in the toxicity of uremic serum. Overall, Berman et al. demonstrated that uremic serum is toxic to zebrafish embryos through a complement-mediated pathway [214].

A study performed by Perna A.F. et al. explored the effects of lanthionine, a pending uremic toxin, on the cardiovascular system of zebrafish larvae [217]. Lanthionine is a non-proteinogenic amino acid generated as a side-product of the action of trans-sulfuration enzymes in hydrogen sulfide (H_2_S) biosynthesis, and it has been demonstrated as a potential uremic toxin with several untoward effects [211,212,213,218]. Lanthionine has been detected in two orders of magnitude concentrations higher in hemodialysis patients compared to normal subjects, where this compound is almost undetectable in circulation [202,203]. In this study, embryos were treated with various concentrations of lanthionine (0.3–2 μM), alone or in combination with glutathione. The redox state has been investigated since the enzymes that catalyze the complete trans-sulfuration pathway are also independently responsible for H_2_S biosynthesis and are influenced by the redox state. The authors explored these compounds at different embryo developmental stages, showing that lanthionine at 0.3 μM (concentration detected in hemodialysis patient blood) induces alteration in cardiac morphogenesis [217]. Specifically, lanthionine, in a dose-dependent way, was able to alter the ventricle/atrium size ratio, evaluated from the 2D imaging. Moreover, 30% of embryos in this experimental group showed yolk sac swelling or deformation. Lanthionine influenced the heartbeat frequency, inducing a dose-dependent susceptibility to arrhythmia compared to the control. When GSH was administered after the start of the heart development (short-lasting effect), the heart rhythm result was more similar to that of the control. Utilizing DanioVision analysis, the authors quantified the locomotor activity of the embryos after treatment, proving that lanthionine induced alterations of locomotor parameters, such as heading and rotation. Perna and their coworkers investigated the molecular alteration induced by lanthionine, showing that this uremic toxin exerted acute effects on trans-sulfuration enzymes, and the expression of genes involved in the inflammation and metabolic regulation, and modified the microRNA expression in a way that was comparable with some alterations detected in uremia [217].

Arinze et al. observed that doses as low as 50–100 μM IS suppressed vasculogenesis, including decreases in the thickness and bifurcation of the intersegmental vessel (ISV) and the tail microvasculature in zebrafish embryos [219].

Additionally, Tang PW et al. investigated the impact of IS on the development of zebrafish embryos, especially on cardiac and renal development [220]. At 24 hpf, zebrafish were exposed to IS at concentrations ranging from 2.5 to 10 mM. IS reduced the survival and hatching rate, and caused cardiac edema and morphological malformations, including an elongated and separated atrium and ventricles. Moreover, after IS exposure for 24 h, the production of endogenous ROS was increased. To evaluate the impact of IS on renal function, the embryos were soaked with 2.5 mM of IS at 24 hpf and injected with 50 ng of 10 kDa dextran conjugated with Rhodamine fluorescent dye into the cardiac venous sinus at 48 hpf, showing abnormal renal development after the exposure to 1.25 mM and 2.5 mM of IS. To investigate the molecular pathway involved in this alteration, the authors demonstrated that IS affected zebrafish development via the ROS and MAPK pathways, which subsequently led to inflammation in the embryos [220].

As indicated, the first work providing the suitability of zebrafish as a model to study uremic toxicity is quite recent. During the last decade, there has been an increasing interest in using zebrafish as a screening model to study human drug disposition and, in general, the kidneys [221]. This increasing interest is also related to the numerous progress seen in the field of technological development for zebrafish analysis, as well as to the increasing knowledge of the uremic toxicity mechanism in humans. Some examples of the techniques for zebrafish implemented in recent years include the 3D high-resolution imaging or the mass screening approach, which provide, respectively, an in vivo image of the whole organism and the toxicological or genetic effect of the compounds [222,223]. In addition, compared to other animal models, such as rodents, the external fertilization of zebrafish makes it possible to observe the normal or abnormal development of the organs or the time-dependent effect of UTs. This is not possible in rodents, where the sacrifice of the mother and embryos is required. These aspects, together with the advantages described in Section 2, are making zebrafish a suitable tool for advancing the UTs cardiotoxicity knowledge in the future.

However, it is also important to underline that some limitations regarding the UTs cardiotoxicity study related to the anatomical differences between zebrafish and humans exist. In fact, studies have shown that UTs affect normal cardiac electrophysiology, with potentially pro-arrhythmogenic effects [224]. Unlike humans, zebrafish have differences in their inward current, where their sodium current is lower and calcium current is higher [225]. In addition, the two-chambered hearts of zebrafish have differences in calcium cycling and, with the absence of sarcolemma T-tubules in cardiomyocytes, this is important for excitation-contraction coupling [225,226]. Studies have also demonstrated that UTs have a pro-fibrotic and pro-hypertrophic effect on cardiac tissue [227]. As a difference to mammalian models, in zebrafish, no chronic fibrosis is detected because of the high regenerative capacity of the heart [228].

## 6. Conclusions

Cardiovascular events are the main mortality cause in the general population and above all in CKD, with a cardiovascular risk 10 times higher in these patients than in normal people. The development of new experimental approaches, which include the use of non-mammalian models, such as zebrafish, allows for a good understanding of the cardiovascular system at both the physiological and molecular levels. The zebrafish and human cardiovascular systems are comparable and the disease-model design of different zebrafish lines is possible because of the zebrafish amenity of genetic manipulation and gene homology compared with humans. These advantages make zebrafish a perfect model to reproduce cardiac defects.

Zebrafish are increasingly used as a model for drug discovery and for assessing compound toxicity, such as UTs. The toxicity reported in zebrafish is representative of other species, including humans, and the study conducted with embryos and larvae is easier compared to rodent models. Despite its incredible advantages, the use of zebrafish to study the effects of UTs on the cardiovascular system has only been employed for a few years.

Although this model cannot completely replace rodents, extensive automated processing techniques are being developed, which will allow the effect of drugs and compounds on every organ. Considering that the effects on zebrafish are comparable to humans, the results will also provide more data regarding some diseases that are still poorly understood.

In this way, the optimization of technologies available today will provide the spreading of zebrafish in the study of uremic toxicity.

## Figures and Tables

**Figure 1 ijms-24-05656-f001:**
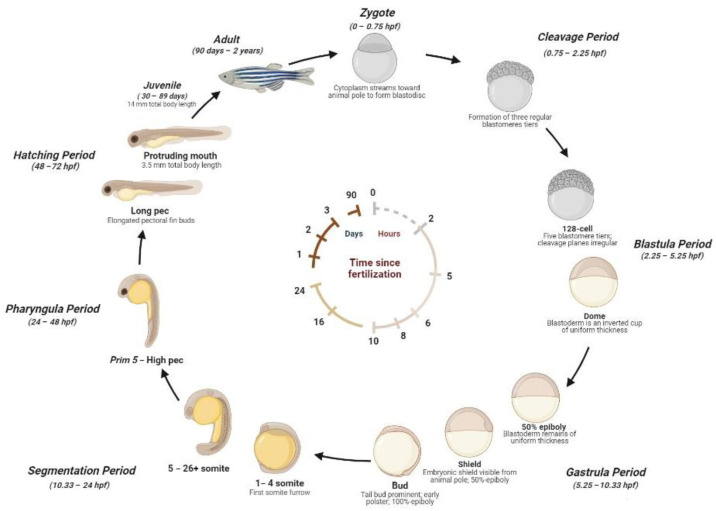
Zebrafish development stages. *Zygote* (1k-cell stage) (0–0.75 hpf): the newly fertilized egg until the first cleavage occurs (after 40 min). Cleavage Period (0.75–2.25 hpf): the formation of three regular tiers of blastomeres occurs. Blastula Period (2.25–5.25 hpf): at the 128-cell stage, five blastomere tiers are formed and the cleavage planes are irregular. At the 1-k-cell stage, a yolk syncytial layer can be present, with an irregular form and asynchronous cell cycle. At 4 hpf, the egg has a spherical shape with a flat border between the blastodisc and yolk. At the dome stage, the yolk cells are bulging toward the animal pole, beginning the epiboly formation around the egg. *Gastrula Period* (5.25–10.33 *hpf*): between 50% of epiboly to the shaping stage, the blastoderm remains of uniform thickness and the embryonic shiels are visible from the animal pole. Between about 60% and 100% of epiboly continues the shape, becoming more along the animal-vegetal axis. The axis and neural plate, brain, and notochord are rudimentary. At the bud stage, the epiboly is completed and the yolk plug is closed, with the formation of a tail bud prominent. *Segmentation* (10.33–24 hpf): at the 1–4 somite stage, the first somitic furrow forms. During this stage, an optic primordium, otic placode, otic vesicle, weak muscular contractions, blood islands, otoliths, and midbrain-hindbrain boundary are present. *Pharyngula Period* (24–48 hpf): At Prim5-High pec, the zebrafish present early pigmentation, heartbeat, early touch reflex, pigmented retina, early motility, and tail pigmentation of the pectoral fins. *Hatching Period* (48–72 hpf): The organs are completed, and the fin, jaw, and gills develop quickly, with a 7.8 mm total body length at the end of this stage. *Juvenile* (30–89 days): The adult fins and pigment are present, reaching 14 mm of total body length. Adult stage (90 days–2 years): The fishes are sexually mature and ready for breeding.

**Figure 2 ijms-24-05656-f002:**
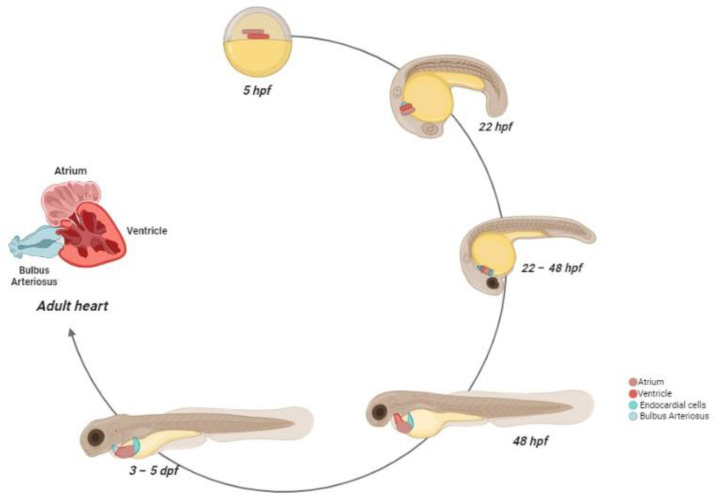
Zebrafish heart development. At 5 hpf, the cardiac progenitors are positioned in the lateral marginal zone. The atrial progenitor cells are located more ventrally than the ventricle progenitors, positioned at the margin site. At 22 hpf, the progenitors express chamber-specific genes, inducing cell structure differentiation. After this process, the cells reach the midline and, after the fusion with the developing endocardial cells, form the cardiac disc, where atrial cardiomyocytes surround the ventricle cells. Between 22–24 hpf, the heart starts beating. At 28 hpf, the cardiac disc elongates, forming a linear heart tube, which, in turn, begins before the leftward migration and then the looping. At 48 hpf, the heart chambers are now formed and easily distinguishable, and the bulbus arteriosus and the sinus venosus are matured as blood outflow and inflow tracts, respectively. At 72 hpf, the trabeculae formations start and rapidly expand in the developing heart. The pacemaker cells are localized in the inner curvature of the atrium. Between the larval and juvenile stage, the atrium and ventricle rotate, so that the ventricle is positioned ventrally to the atrium. At the *adulthood* stage, the coronary arteries feed the ventricle and a compact myocardium is formed.

**Table 1 ijms-24-05656-t001:** Massry/Chock’s requirements for uremic toxin classification.

Massry/Chock’s Requirements
Toxin must be chemically identified and characterized
Toxin must be quantified in the biological fluids
Toxin in fluids must be detected at high levels in the uremia
Relation between the toxin levels and one or more manifestations of the uremia
Reduction in the level of the toxin must be correlated with an improvement in uremic effects
Uremic toxin levels present in the uremia must be used in vitro and in vivo models to reproduce uremic manifestation A possible pathobiological mechanism should be demonstrated to explain the link between the toxin and uremic manifestation

**Table 2 ijms-24-05656-t002:** Physicochemical characteristics for uremic toxin classification.

Classification	Characteristic	Example
Free water-soluble low-molecular weight compounds	<500 Da; removed by dialysis	Creatinine; urea; uric acid; and Trimethylamine
Medium compounds	>500 Da; removed by dialysis membranes with large pores	Cystatin C; Cytokines; FGF23; β2-microglobulin; and PTH
Protein-bound compounds	Non-dialysable	Indoxyl Sulfate; P-Cresol; Indole-2-acetate; Homocysteine; and Advanced Glycation end Products

## Data Availability

Not applicable.

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
