# Peer review of "Zebrafish as a Model of Cardiac Pathology and Toxicity: Spotlight on Uremic Toxins"

_ijms, 2023, doi:10.3390/ijms24065656_

Round 1

Reviewer 1 Report

Coppola, A., Lombari, P., Mazzella, E., Capolongo, G., Simeoni, M., Perna, A.F., Ingrosso, D., Borriello, M. (2023) Zebrafish as a model of cardiac pathology and toxicity: spotlight on uremic toxins. Int. J. Mol. Sci. 24: submitted.

Summary: This is a well written manuscript review. The authors have done a nice job of introducing the Zebrafish model and highlight its value and advantages in the study of cardiac disease and chronic kidney disease. The model is well introduced and explained for those marginally familiar with the system. However, it has some oversights which require more work. The description of the model system is not complete in that the limitations are not discussed. In addition, when original research is described not enough detail or interpretation as to the key role the Zebrafish model has played is written.  

Major Points:

Ø  While the manuscript covers some of the benefits of using Zebrafish as a model system, to be complete and unbiased the disadvantages of the model must be addressed as well. It could take the form of a section addressing advantages and disadvantages. Alternatively, it could be addressed when appropriate in the text.

Ø  Since the review focusses on uremic toxins, at a minimum, these sections need more detail. The authors have carefully defined uremic toxins in section 4, but it does not seem to specifically relate to Zebrafish. They address this is section 5. It would be more interesting to have a more concise section 4 and expanded section 5. Additional information which could be provided in section 5 relate to how the authors believe the Zebrafish model has answered questions which could not be addressed or answered as well in other models. How the work they are describing is important and seminal to the field. Where the Zebrafish model is currently as compared to 5-10 years ago in the field. As the focus of the review, this section should make clear the role of Zebrafish in current research and it continued value for the future. The authors could address this point by different means, however the end result should be a clear picture of how the Zebrafish model has been instrumental in advancing uremic toxins and cardiovascular disease.

Minor Points:

Ø  There are too many subject reviews in the references. The authors should focus on the research articles and highlight those where there have been great advances in the field using the Zebrafish model.

Ø  I am not sure “impetuous” within line 118 is the correct word for what the authors were trying to say as it generally implies a degree of carelessness or imprecise thought in the action. If this was the intended meaning it is fine.

Ø  In the legend of Figure 1, “Zigote” should be changed to Zygote. It seems advantageous to put “1-k-cell stage” in more general terms or define in parentheses.

Author Response

We thank the reviewer for the general evaluation of the paper. We much appreciated the Reviewer’s comment, as it strongly enriches our manuscript.

Major Points:

  • While the manuscript covers some of the benefits of using Zebrafish as a model system, to be complete and unbiased the disadvantages of the model must be addressed as well. It could take the form of a section addressing advantages and disadvantages. Alternatively, it could be addressed when appropriate in the text.

Reply: Thank the reviewer for the comment. We added zebrafish disadvantages at section 2, line 111-117.

Specifically, we added the following sentences: “Like all animal models, also zebrafish presents some disadvantages. Lack of maternal-embryo interaction during gestation and the presence, up to 48-72 hpf, of the chorion, may create problems for drug permeability. The extremely rapid development could be a disadvantage, where the screening in zebrafish appears as a running-away train, requiring a perfect setting of experiments to have the right exposure time. Many genes in zebrafish are orthologs to human, but are present in two copies, creating another molecular pathway with different effects than the main one [23].”

  • Since the review focusses on uremic toxins, at a minimum, these sections need more detail. The authors have carefully defined uremic toxins in section 4, but it does not seem to specifically relate to Zebrafish. They address this is section 5. It would be more interesting to have a more concise section 4 and expanded section 5. Additional information which could be provided in section 5 relate to how the authors believe the Zebrafish model has answered questions which could not be addressed or answered as well in other models. How the work they are describing is important and seminal to the field. Where the Zebrafish model is currently as compared to 5-10 years ago in the field. As the focus of the review, this section should make clear the role of Zebrafish in current research and it continued value for the future. The authors could address this point by different means, however the end result should be a clear picture of how the Zebrafish model has been instrumental in advancing uremic toxins and cardiovascular disease.
  • Reply: We thank the Reviewer for the comment. As the Reviewer affirmed, the section 4 is not specifically related to Zebrafish. Indeed, the aim of this section is to give to the readers an exhaustive vision about UTs. We hope that we successfully addressed this aim.
  • Accordingly with the Reviewer’s comment, we worked on the section 5, provided informations about the reason why zebrafish represents a suitable model to study uremic toxicity. Moreover, we better explain in the text that zebrafish is used for this kind of study only from the last ten years. The reasons are expecially related to the knowledge increase in the role of UT sas a driver of CVD in CKD, concomitantly with the technological development allowing a more accurate zebrafish cardiac analysis. Specifically, to explain these, we added the sentences “Despite the numerous progresses related to the development of tools for recording and analyze various zebrafish parameters, only in recent years, zebrafish embryo has been used as a model to evaluate the uremic toxicity. The first work, in this field, was provided by Berman et al in 2013 [214].” at the lines 645-657, and “As indicated, the first work providing the suitability of zebrafish as model to study uremic toxicity is quite recent. During the last decade, there has been an increasing interest in using zebrafish as a screening model to study human drug disposition and, in general, kidney [222]. This increasing interest is also related to the numerous progresses done in the field of technological development for zebrafish analisys, as well as to the increasing knowledge in uremic toxicity mechanism in huma Some examples of techniques for zebrafish implemented in recent years include the 3D high resolution imaging or the mass screening approach, which provide, respectively, an in vivo image of the whole organism and toxicological or genetic effect of compounds [222, 223]. In addition, compared to other animal models, such as rodents, external fertilization of zebrafish makes it possible to observe normal or abnormal development of organs or the time-depended effect of UTs. This is not possible in rodents, where sacrifice of the mother and embryos is required. These aspects, together with the advantages here described, in the Section 2, are making zebrafish a suitable tool in advancing of UTs cardiotoxicity knowledge in the future.” At the lines 714- 727.
  • To give an comprensive vision about the use of zebrafish in the study of uremic toxicity, we have now added in the lines 728-738 the limitations related to this model in this specific field.

We strongly think that this Reviewer’s comments increase the quality of our manuscript

Minor Points:

  • There are too many subject reviews in the references. The authors should focus on the research articles and highlight those where there have been great advances in the field using the Zebrafish model.

We Thank the reviewer for this comment. We have replaced most of the reviews with original articles relevant to the topics treated. All substitutions are marked in the References Section.

  • I am not sure “impetuous” within line 118 is the correct word for what the authors were trying to say as it generally implies a degree of carelessness or imprecise thought in the action. If this was the intended meaning it is fine.

Thank the reviewer for this note. “Impetuous” was referring to the speed with which zebrafish has been increasingly used in research over the years. We modified “impetuous” with “increasingly popular”, at line 124.

  • In the legend of Figure 1, “Zigote” should be changed to Zygote. It seems advantageous to put “1-k-cell stage” in more general terms or define in parentheses.

We thank the reviewer for this clarification. We have corrected and added this detail to the legend of the Figure 1, at line 145.

Reviewer 2 Report

1. It is suggested to add the possible future research directions with the zebrafish model. 2. Discussing the limitations and possible problems in the application of this animal model in the uremic toxin study.

Author Response

We thank the reviewer for the general evaluation of the paper. We much appreciated the Reviewer’s comment, as it strongly enriches our manuscript.

  • It is suggested to add the possible future research directions with the zebrafish model.

We Thank the Reviewer for this suggestion. We modify the text of the section 5, by adding the sentences “As indicated, the first work providing the suitability of zebrafish as model to study uremic toxicity is quite recent. During the last decade, there has been an increasing interest in using zebrafish as a screening model to study human drug disposition and, in general, kidney [221]. This increasing interest is also related to the numerous progresses done in the field of technological development for zebrafish analisys, as well as to the increasing knowledge in uremic toxicity mechanism in human. Some examples of techniques for zebrafish implemented in recent years include the 3D high resolution imaging or the mass screening approach, which provide, respectively, an in vivo image of the whole organism and toxicological or genetic effect of compounds [222, 223]. In addition, compared to other animal models, such as rodents, external fertilization of zebrafish makes it possible to observe normal or abnormal development of organs or the time-depended effect of UTs. This is not possible in rodents, where sacrifice of the mother and embryos is required. These aspects, together with the advantages here described, in the Section 2, are making zebrafish a suitable tool in advancing of UTs cardiotoxicity knowledge in the future.” at the lines 714-727.

  • Discussing the limitations and possible problems in the application of this animal model in the uremic toxin study.

We Thank the Reviewer for this suggestion. We have added general zebrafish disadvantages as experimental model in the Section 2, lines 111-117 .(“Like all animal models, also zebrafish presents some disadvantages. Lack of maternal-embryo interaction during gestation and the presence, up to 48-72 hpf, of the chorion, may create problems for drug permeability. The extremely rapid development could be a disadvantage, where the screening in zebrafish appears as a running-away train, requiring a perfect setting of experiments to have the right exposure time. Many genes in zebrafish are orthologs to human, but are present in two copies, creating another molecular pathway with different effects than the main one [23]”)

In addition, we have included limitations in the application of zebrafish in the uremic toxins cardiotoxicity study in the Section 5, at the lines 728-738, by adding the sentences “However, it is also important to underline that some limitations regard UTs cardiotoxicity study related to anatomical differences between zebrafish and humans exist. In fact, studies have shown that UTs affect the normal cardiac electrophysiology, with potentially pro-arrhythmogenic effect [224]. Unlike humans, zebrafish has differences in inward current, where sodium current is lower and calcium current higher [225]. In addition, the two-chambered hearts of zebrafish have differences in calcium cycling and with the absence of sarcolemma T-tubules in cardiomyocytes, important for excitation-contraction coupling [225, 226]. Studied have also demonstrated that UTs have pro-fibrotic and pro-hypertrophic effect on cardiac tissue [227]. As a difference to mammalian models, in zebrafish no chronic fibrosis is detected, because of the high regenerative capacity of the heart [228].”